# Advances in Modification Methods Based on Biodegradable Membranes in Guided Bone/Tissue Regeneration: A Review

**DOI:** 10.3390/polym14050871

**Published:** 2022-02-23

**Authors:** Yue Gao, Shuai Wang, Biying Shi, Yuxuan Wang, Yimeng Chen, Xuanyi Wang, Eui-Seok Lee, Heng-Bo Jiang

**Affiliations:** 1The CONVERSATIONALIST Club, School of Stomatology, Shandong First Medical University & Shandong Academy of Medical Sciences, Tai’an 271016, China; yuegaoo@outlook.com (Y.G.); wangs0227@outlook.com (S.W.); sgs0525@outlook.com (B.S.); wangaxuan2020@outlook.com (Y.W.); wysbd7474@outlook.com (Y.C.); xuan05184@outlook.com (X.W.); 2Department of Oral and Maxillofacial Surgery, Graduate School of Clinical Dentistry, Korea University, Seoul 08308, Korea

**Keywords:** guided tissue regeneration, guided bone regeneration, biodegradable polymer, membrane, material modification

## Abstract

Guided tissue/bone regeneration (GTR/GBR) is commonly applied in dentistry to aid in the regeneration of bone/tissue at a defective location, where the assistive material eventually degrades to be substituted with newly produced tissue. Membranes separate the rapidly propagating soft tissue from the slow-growing bone tissue for optimal tissue regeneration results. A broad membrane exposure area, biocompatibility, hardness, ductility, cell occlusion, membrane void ratio, tissue integration, and clinical manageability are essential functional properties of a GTR/GBR membrane, although no single modern membrane conforms to all of the necessary characteristics. This review considers ongoing bone/tissue regeneration engineering research and the GTR/GBR materials described in this review fulfill all of the basic ISO requirements for human use, as determined through risk analysis and rigorous testing. Novel modified materials are in the early stages of development and could be classified as synthetic polymer membranes, biological extraction synthetic polymer membranes, or metal membranes. Cell attachment, proliferation, and subsequent tissue development are influenced by the physical features of GTR/GBR membrane materials, including pore size, porosity, and mechanical strength. According to the latest advances, key attributes of nanofillers introduced into a polymer matrix include suitable surface area, better mechanical capacity, and stability, which enhances cell adhesion, proliferation, and differentiation. Therefore, it is essential to construct a bionic membrane that satisfies the requirements for the mechanical barrier, the degradation rate, osteogenesis, and clinical operability.

## 1. Introduction

Periodontitis is a bacterial infection-induced chronic inflammation that is associated with enhanced neutrophil and macrophage infiltration, as well as osteoclast activation via RANKL signaling [1]. It can lead to severe periodontal disease and the loss of alveolar bone. Periodontal tissue, as well as alveolar bone and cementum, are part of the periodontal region: a unit consisting of numerous tissues surrounding and functionally supporting teeth. Periodontal disease damages the paradentium, which can lead to the loosening of teeth [2]. Extant treatments for periodontitis mainly target the symptoms, including the removal of plaque and reducing inflammation [3,4]. Though these therapies retard disease progression, they do not address the reattachment of periodontal tissue to the tooth or the restoration of the periodontal tissue; thus, dental function remains inhibited [5,6]. An ideal approach to treating periodontitis is to reestablish the complex hierarchical structure of the periodontal tissue, which includes fresh cementum, alveolar bone, as well as periodontal ligament and gingival tissue. The restoration of the periodontal tissue by guided bone/tissue regeneration is an increasingly important challenge faced by clinicians [1].

A basic requirement for guided tissue/bone regeneration (GTR/GBR) is the introduction of an effective physical barrier between soft tissue and bone tissue (Figure 1) that addresses the distinct migration speeds of periodontal tissue cells (in descending order of growth rate, gingival epithelial > gingival connective tissue cells > periodontal membrane cells > the alveolar bone cells). The membrane should prevent gingival soft tissue from accessing the bone defect area before occupying the root surface, allowing pre-dental cells to occupy the next surface and differentiate into cementum cells, fibroblasts, and osteoblasts, promoting the healing process [7,8]. Bone graft materials are often used in conjunction with the membrane in bone deficient areas, serving as a scaffold for new bone formation [9].

An ideal GTR/GBR membrane should include a large membrane exposure area, biocompatibility, hardness, ductility, a membrane void ratio, clinical manageability, along with allowing for tissue integration and cell occlusion, though no such membrane is currently available for therapeutic application. The GTR/GBR materials described in this study meet all of the fundamental ISO requirements for human use, as determined through risk analysis and rigorous testing. The standard we mentioned is ISO 22803.

According to their degradation capacity, GTR/GBR membranes are classified as absorbable and nonabsorbable. Nonabsorbable membranes were mainly developed for the maintenance of postoperative clearance; they have mechanical properties that are suitable for bone defect reconstruction and they inhibit cell migration [11,12]. Generally, nonabsorbable membranes include titanium mesh (Ti mesh), polytetrafluoroethylene (PTFE), and titanium-reinforced PTFE membranes. Based on the structure, PTFE can further be classified into expansion-PTFE (e-PTFE) and high-density-PTFE (d-PTFE). e-PTFE, developed in the 1990s, is considered a standard material for clinical applications, although it suffers from high porosity and exposure [13]. The d-PTFE membranes have a comparatively smaller pore size (less than 0.3 mm), which inhibits the integration of tissue into the membrane and reduces the difficulty of membrane removal [14,15]. The greatest disadvantage of nonabsorbable membranes is that they require a secondary surgery for removal, which interferes with the healing process and increases the risk of bacterial infection.

Absorbable membranes can be made from natural or synthetic materials. Natural absorbable materials include collagen, chitosan, and gelatin [16]. The most common synthetic absorbable materials include organic aliphatic thermoplastic polymers such as polylactic acid (PLA), polyglycolic acid, and their copolymers [9]. They have high biocompatibility, low mechanical strength, and promote tissue healing. Because they are absorbed and most have antibacterial properties, they reduce the probability of a second operation or other clinical intervention. However, the degradation process may affect the surrounding tissues and even cause oral diseases. The main challenge of bioabsorbable membranes is the synchronization of the absorption time with the tissue formation cycle. The membrane degradation rate cannot be controlled, and the membrane’s low mechanical strength can cause a loss of spatial support [17].

Although both absorbable and nonabsorbable membranes are superior to internal applications, GTR/GBR membranes still require a considerable amount of research for the development of ideal materials. Modern research of GTR/GBR membranes considers structural (e.g., cell compatibility and degradability) as well as functional properties, such as the incorporation of growth factors and antibiotics with chemically controlled release times [11,18,19]. Although most of the research has occurred in vitro, the collective findings of these studies reflect the immense potential for clinical application. The continuous clinical development of GTR/GBR membranes necessitates a comprehensive review of promising membrane modifications. In a review of related literature on *PubMed*, *Wiley*, *Cochrane*, *Embase*, and *ScienceDirect*, we found very few resources related to modified nonabsorbable membranes over the past five years. Due to the practical limitations of nonabsorbable membranes, most modern studies focus on absorbable membranes. Based on this, the present review was focused on the improvements made in absorbable membrane technology during the last five years.

Many investigations have attempted to modify the original membrane technology to create the ideal membrane. In recent years, nanomaterials have become a major topic of study in bioengineering. They have been used in directing bone/tissue regeneration and exhibit excellent cell adhesion and bone regeneration properties. Nanofibers are the most commonly used polymer structures in tissue engineering. Their microscopic particles and intrinsic features allow for an excellent barrier membrane effect, prohibiting cell transmission through the membrane [20]. Membranes, or scaffolds, are generated with a high surface-to-volume ratio that can incorporate and release proteins, medicines, and ligands. Because the fibers may be customized in size, orientation, filling, porosity, and density, the mechanical and morphological features of these membranes are more reliable within the desired context. Finally, the morphology of the extracellular matrix (ECM) is replicated by the three-dimensional structure, which is mostly composed of collagen fibrils with elastin and other macromolecules. Nanofibers can also influence stem cell activity and enhance certain cellular processes including adhesion, proliferation, and differentiation [21]. Electrospinning (ELS), which uses a polymer solution in a high-electric field, is the most commonly employed technology for fabricating nanomaterials. ELS technology has grown in popularity since its inception in the 1930s and has seen numerous advancements in its fundamental components and applications [22,23]. ELS is one of the most successful membrane production techniques and can generate nanoscale fibers that promote the reestablishment of the natural ECM [24]. The traditional fiber membrane has a two-dimensional structure, while the nanofiber membrane prepared by ELS has a three-dimensional structure, which allows for the incorporation of various useful properties with a simple and relatively inexpensive production method. The GTR/GBR scaffold is composed of a loose, porous solution electrospinning writing (SEW) layer, which supports and promotes bone growth, and a dense solution electrospinning (SES) layer, which is resistant to interference by non-osteoblasts [25]. However, the dense arrangement and lamellar assembly of electrospinning membranes make it suitable only in the construction of dense layers and the performance of barrier functions. The layer-by-layer manner of assembly allows for the generation of three-dimensional porous scaffolds with customizable properties, which has, in recent years, sparked interest in 3D printing for tissue-engineered scaffolds [26,27]. One of the most extensively used functional components of this process is small molecule osteogenic medicines, or antibiotics [28,29,30].

The involvement of bioactive ions in antibacterial activity, tissue repair, and immunological modulation, among other processes, has received a great deal of attention in recent years. Despite substantial advancements in membrane design, no effective electrospun GTR/GBR membrane is currently available for clinical application. This paper classifies GTR/GBR membranes into synthetic polymer membranes, natural polymer membranes, and metallic membranes, according to their primary materials; reviews the modification methods for different membranes, including new membrane manufacturing techniques, surface modifications, and changes in membrane composition; and analyzes the effects of various methods on membrane properties. The ultimate aim of this study is to provide a reference for developing ideal GTR/GBR membranes.

## 2. Materials and Methods

In this review, only the literature published in English is included, and the last search was conducted in August 2021 (Figure 2).

## 3. Results and Discussion

A total of 965 articles were identified using an electronic database and a manual search; 68 articles met the inclusion criteria (Figure 2). Based on these articles, we divided the barrier films into three categories (a total of nine subcategories) to discuss their material advantages, disadvantages, and modification methods (Figure 3).

### 3.1. Resorbable Membranes Based on a Synthetic Polymer

#### 3.1.1. Polycaprolactone (PCL)

Polycaprolactone (PCL) is a biomedical synthetic polymer with suitable mechanical properties and biocompatibility, good solubility in most organic solvents, and which is easily processable, making it popular in the field of bone/tissue repair [31,32,33,34,35,36]. However, its clinical application is limited by its slow degradation rates, poor osteoconductivity, and low bioactivity [37,38,39]. To address these limitations, other materials can be combined with PCL to form special membrane structures, or functional groups can be attached to improve its clinical efficacy [7]. Additional functional properties can also be incorporated into PCL by loading passively released drugs [33,37,40]. The various properties of modified PCL membranes are shown in Table 1.

With the addition of other components, specialized structures can be generated to achieve enhanced results. For example, a core-shell structure is generated by an ordered assembly of nanomaterials that are attached through chemical/electrical bonds (Figure 4). Wang et al. used core-shell structures to integrate the properties of both internal and external materials. The core of the nanofiber was gelatin and metronidazole (MNA), while the shell was composed of PCL and nano-hydroxyapatite (NHA). MNA imparted antimicrobial properties to PCL, while NHA promoted osteogenesis [41]. In another study comparing the incorporation of different amounts of bioactive glass (BG), PCL + 2%, BG showed higher cell adhesion rates and cell survival, as well as enhanced fiber and pore diameter, all of which are conducive for GTR membranes [42]. Another specialized structure, a nanopattern (Figure 5), promoted cell growth on the membrane surface. Jang et al. utilized a combination of equine bone powders (EBPs) and nanopatterning to modify the PCL membranes. The addition of EBPs improved the hydrophilicity of the PCL membranes and facilitated cell diffusion and aggregation, while the nano grooves facilitated cell elongation. The growth and differentiation of human dental pulp stem cells (DPSCs) were enhanced by the synergistic effects of EBPs and nanopatterning; this is considered an effective method to boost the growth capacity of cells [43].

One drawback of PCL is its relatively low level of bioactivity that inhibits cell adhesion and growth. BG is hydrophilic and its incorporation in the PCL membrane enhances cell adhesion, elongation, and proliferation [34]. Membranes incorporating 7 wt% copper-free BGs have shown good surface wettability and osteogenic ability while also facilitating cell proliferation and the adhesion of adipose-derived stem cells (ADSCs) [34]. A new bioactive glass material, F18, exhibits excellent stability, bioactivity, and antibacterial properties and promotes vascular tissue and new bone growth, making it ideal for dental and orthopedic applications [36]. The incorporation of BGs also enhanced the mechanical properties of the PCL membrane [34,36,46]. Multiple BGs can also function collectively to enhance osteogenesis. In a study by Terzopoulou et al., two types of mesoporous BGs (SiO_2_-CaO-P_2_O_5_ and SiO_2_-SrO-P_2_O_5_), compounded with PCL by spin-coating, increased the hydrophilicity and bioactivity of the original membrane. Loading the PCL membrane with the bisphosphonate drug ibandronate, together with Sr in BG, also enhanced osteogenesis [39].

Silicate ions and calcium ions play a crucial role in the formation of hydroxyapatite (HA), an essential component of bone, and can enhance the osteoconductivity of the PCL membranes. The silicate ion is the initial nucleation point for HA crystal growth, and calcium ions accelerate HA crystal growth. An HA coating was prepared on membranes composed of PCL and CaO-SiO_2_ gel fibers. CaO-SiO_2_ released silicate ions and calcium ions, and the HA coating significantly enhanced the osteoconductivity and osteogenesis of the GBR membranes [45]. According to Ezati et al., a similar effect can be achieved by adding different ratios of tricalcium phosphate (β-TCP) to PCL/gelatin/chitosan. β-TCP, like CaO-SiO_2_ gel fibers, can act as a precursor for Ca^2+^ and PO_4_^3-^ to facilitate osseointegration. Its degradation products neutralize changes in pH during degradation; thus, by adjusting its content levels, the rate of membrane degradation can also be regulated. The mechanical properties, wettability, and roughness of the membranes were all enhanced with increased levels of β-TCP [48]. Interestingly, sterilization of PCL membranes with low-temperature hydrogen peroxide gas plasma (LTP) not only promotes suitable cell morphologies, but also allows for better osteogenic differentiation of pre-osteoblasts [44].

As a antimicrobial drug which plays an important role in treatment modification, strontium-substituted hydroxyapatite nanofibers (SrHANFs) have excellent drug loading capacity and the potential to release antimicrobial drugs. Tsai et al. also found that the addition of SrHANFs to the PCL membranes promoted osteoblast mineralization and differentiation [49]. However, the rapid release rate of antibiotics from SrHANFs inhibits the overall antibacterial activity and emphasizes the importance of controlled release rates in the development of optimal GTR/GBR membranes [7]. Shi et al. coated the PCL surface with polydopamine and grafted metronidazole (MNA) onto the surface through ester bonds; here, the release rate of MNA was determined by the concentration of cholesterol esterase (CE) at the site of infection. Moreover, the rate of drug release depends on the type of chemical bond that connects the drug to the nanofiber surface [32]. In a previous study by the same authors, the rate of drug release was regulated by changing the ratio of gelatin to drug, which is a relevant factor in explaining the rate of drug release from a macroscopic point of view [32]. However, the incorporation of certain antimicrobial drugs may have side effects. The incorporation of hydroxyapatite nanoparticles (HANPs) can promote cell differentiation while imparting antimicrobial properties, but the amount of HANPs can drastically alter the membrane’s mechanical properties: 10% particle incorporation improves the tensile properties of the membrane, while 20% leads to a decrease in these properties [33]. Münchow et al. found that, although it improved antibacterial activity, cell proliferation, and wound healing, ZnO at ≥5 wt% weakened the mechanical properties, and the addition of 30% ZnO actually decreased the biocompatibility of the membranes [40]. Therefore, it is particularly important to select and add appropriate amounts of reagents to maintain/improve the structural properties of the PCL membranes, while imparting a controlled drug release rate.

To conclude, choosing a variety of suitable materials in combination with the PCL membranes may be the optimal modification method to remediate its multiple limitations and make PCL membranes more suitable for GTR/GBR applications.

#### 3.1.2. Polylactic Acid (PLA)

PLA is a biosafe neotype synthetic polymer with a low degradation rate that degrades to produce carbon dioxide and water. PLA is commonly used to prepare scaffolds for tissue regeneration, especially by electrostatic spinning [50,51]. However, PLA is limited by its comparatively low mechanical strength. Researchers have developed various improvements to the structural properties of PLA, such as the modification of the polymer structure, the preparation of a polymer fiber blend, and the addition of nanoparticles [52,53]. The PLA membranes are frequently created using the solvent casting method [54]. Eventually, 3D printing technology can be used to accurately determine the porosity of the PLA membrane [55]. The incorporation of inorganic particles, such as bioceramics or BG, can also improve the biological activity of the polymer [56]. Antibiotics and carbon nanotubes (CNT) are also added to improve the antibacterial function of the PLA membrane [57]. The United States Food and Drug Administration (USFDA) has approved the use of PLA in biomedical engineering applications [58,59].

Incorporation of nanoparticles is an effective way to modify the GTR/GBR membrane. Abdelaziz et al. studied PLA/cellulose acetate (CA), or PCL nanofiber scaffolds made using electrospinning techniques [33]. Adding different green-synthesized, silver nanoparticles (AgNPs) improved the antibacterial performance and bone regeneration activity of the membrane. Furthermore, the addition of HANPs to the nanofibrous scaffolds improved cell viability by ~50% and enhanced the tensile properties of the scaffold at concentrations of ~10 wt%, but decreased tensile strength at 20 wt% concentrations.

Composite fibrous electrospinning membranes based, on PLA and PCL with borate bioactive glass (BBG), were prepared by Rowe et al., who characterized the membranes using scanning and transmission electron microscopies [56]. After 7 days, the cell proliferation rates of the preosteoblast cells containing BBG were higher than that of membranes without BBG.

Moura et al. incorporated CNT and BG in the PLA membranes and found that the addition of 5 wt% BG improved the surface porosity and bioactivity of PLA [57]. According to the agar diffusion method, CNT showed some antibacterial activity on the membrane. In vitro experiments revealed that this porous membrane was not toxic to cells and allowed cell differentiation. Moreover, the addition of BG and CNT can change the pore shape of the membrane from spherical to irregular. CNT promoted microbial activity and had a synergistic effect with porous PLA, especially in PLA/5BG/1.0CNT.

The performance of the GTR/GBR membrane can also be improved by the fabrication process of the PLA membrane. A bioresorbable polylactide membrane was made by Zhang et al. using 3D printing and was compared with a membrane made by the conventional solvent casting method [55]. The 3D-printed membranes performed better than the solvent cast membranes. A preosteoblast culture experiment, which assessed the performance of 3D-printed membranes with various pore sizes, showed that cell growth was not affected by variation in pore size. In addition, the PLA membranes with various pore sizes prepared by 3D printing had diverse mechanical properties that could be suitable in a wide range of medical applications.

Another modification method consists of fitting the PLA membrane with a metal material. Du et al. fabricated an absorbable magnesium-enhanced PLA-integrated membrane with a coated or bare magnesium AZ91 reinforcement core [60]. The membrane showed good cell affinity, mechanical properties, corrosion resistance, and appropriate degradation rates and exhibited excellent potential for application as a bioresorbable GTR/GBR membrane.

The performance of the PLA biomembrane can be improved by optimizing the production process and/or adding the appropriate nanoparticles/reagents. Overall, PLA shows great potential as a material for GTR/GBR membranes.

#### 3.1.3. Polylactic-Co-Glycolic Acid (PLGA)

Polylactic-co-glycolic acid (PLGA) is a non-toxic polymer comprising hydroxyacetic acid monomers and lactic acid monomers that shows remarkable bio-absorbability, similar to that of collagen membranes, and excellent compatibility. PLGA is widely used in biomedical engineering, pharmaceuticals, and modern industrial practices [61]. The PLGA membrane shows exceptional cytocompatibility, bioactivity, and physicochemical properties with the variation of the proportions of PLA and PGA, altering the methods of polymerization [62]. However, on its own, PLGA is not a suitable material for GTR/GBR membrane applications. Numerous studies have assessed the effects of various additions and technologies on the improvement of cell activity, antibacterial performance, and degradation rates, among others properties [63,64].

The degradability and mechanical properties of the GTR/GBR membrane have received a great deal of attention. Certain particles have been added to PLGA by electrostatic spinning for specific functions, for instance, to effectively alter cell infiltration rates [65]. Higuchi et al. found that ultrasonic HA-coated membranes can retard biodegradation, promote wettability, and release calcium ions to neutralize acidification, even at comparatively high levels of metabolic activity [66].

Modified PLGA membranes are still in the experimental stage and have only been applied in in vitro and animal model studies. Santos et al. described the use of HA in GBR membranes for increased bioactivity and bone conduction and found that HA improved osteoblast size, diffusion, and migration to the membrane. The HA:TCP ratio had varying degrees of effect on the fiber diameter and the crystalline structure and could be manipulated to maximize the impact of the incorporated contents. By carefully adjusting the HA:TCP ratio to 60:40, they could effectively enhance the membrane characteristics and reconstruct the architecture of the bone [10,67].

To enhance bone regeneration and suppress microorganism proliferation, He et al. incorporated internal and external medicines with the membrane. On the other hand, this incorporation will reduce tensile strength by varying drug release durations to increase entirety efficiency [68]. Jin et al. studied the effects of fish collagen (FC) and HA on the tensile strength and degrading behavior of PLGA membranes with different proportions of LA and GA (Figure 6). GTR/GBR membranes have the potential for various applications due to their generally high degree of biosafety and bioactivity [69]. Additionally, a novel bi-layer scaffold structure can be modified to add several functions. Lian et al. created a new type of GBR scaffold by extending the conception of “membrane”, which means a two-dimension struction, to “scaffold”, which means a three-dimensional structure, to develop numerous structures with enhanced osteogenic and antibacterial capabilities. A loosened and cellular SEW layer was used to assist and facilitate bone ingrowth, while a dense and close SES layer was used to oppose non-osteoblast interference. The two-layer membrane can easily be generated and used in clinical practice because of its enhanced mechanical characteristics and biodegradability [70].

Although PLGA is not widely used, it shows great potential for GTR/GBR membrane modification and can be applied to higher clinical standards with more treatments. The addition of CaP is one of the most promising ways to enhance the properties of the membrane.

#### 3.1.4. Other Materials and Improvements

In addition to the common membrane types, researchers are constantly developing novel membrane types. Frequent improvements include developing new materials, adding one or several cytokines, or the use of other methods that improve the functionality of membrane materials. This also includes the verification of the structural and functional properties and the clinical potential of these membranes in in vivo and in vitro experiments. In addition, considering the importance of nanomaterials, we will expand on the introduction of these substances in Section 3.1.4.3.

##### 3.1.4.1. Developing New Materials

There are two main schools of thought for the development of new materials. One is to form new membranes by changing the proportion of common chemicals and controlling the chemical reactions. Pajoumshariati et al. synthesized a copolyester-polybutadiene succinate-glycolate (PBSGL) based on polybutylene succinate (PBS) and polyglycolic acid (PGA) by esterification of a diol (bis[4-hydroxybutyl] succinate, BHBS) and a di-acid (polyglycolic acid, PGL) with different glycolic acid ratios. This new copolyester combined the biocompatibility of PGA with the excellent mechanical properties of PBS. The same authors also evaluated the effect of the glycolic acid ratio on the biocompatibility and osteogenic differentiation of mesenchymal stem cells at a histological level and found that a high glycolic acid ratio was beneficial to bone formation without adverse inflammatory reactions [71].

The combination of multiple preparation methods to form multilayers with multiple characteristics is another promising idea. Wang et al. prepared a gelatin methacrylamine (GelMA)/poly (ethylene glycol) diacrylate (PEGDA) fiber membrane, a bilayer structure with fiber nanostructure and hydrogel properties, through electrostatic spinning and photocrosslinking [72]. The mechanical strength and degradation time of GelMA fiber membranes were superior to that of crosslinked GelMA fiber membranes. High porosity at the bottom of the membrane favored the adhesion and proliferation of osteoblasts, while the low porosity at the top of the membrane prevented the migration of fibroblasts to the bone defect area. Controlling cross-linking time and PEGDA can also modulate physical properties, degradation rate, cell adhesion, and proliferation [73].

##### 3.1.4.2. Addition of a Functional Substance

Common additives include proteins, cytokines, or nanoparticles that promote bone regeneration. Bone morphogenetic protein (BMP), which belongs to the TGF-β family, is a group of highly conserved functional proteins with similar structures. BMP can stimulate DNA synthesis and cell replication, thus promoting the directed differentiation of mesenchymal cells into osteoblasts. A membrane has been developed with a novel bone graft drug delivery coating composed of biomimetic calcium phosphate (BCAP) layered with a polylysine/polyglutamic acid polyelectrolyte multilayer (PEM). BMP-2 is added to the coating, and BCAP PEM is then deposited. Fibroblast growth factor-2 (FGF-2), which will lead to immediate cell reaction, is adsorbed into the PEM layer and BCAP PEM temporarily delays the cell response to BMP-2. The BMP-2-coated and FGF-2-coated scaffolds were implanted into mouse skull models. One week later, low-dose FGF-2 and BMP-2 showed cell proliferation, including Sca-1^+^ progenitor cells. The addition of the BCAP layer in PEM delays the entry of BMP-2 and allows FGF-2-stimulated progenitor cells to fill the scaffold before the differentiation of BMP-2. To improve bone defect healing, bone mesenchymal stem cells (BMSCs) were pretreated with bFGF and BMP-2 [74]. The results showed that bFGF promoted the proliferation of BMSCs better than BMP-2. The process of bone differentiation induced by the BMP and FGF acellular dermal matrix (ADM) membrane was compared in a critical size defect model, which showed that bFGF-ADM was more effective in recruiting BMSCs [75].

Compared with BMP, nanoparticles play a greater role in bone regeneration at the cellular level, mainly enhancing the mechanical properties and antibacterial ability of the membrane. Kouhi et al. added fibrinogen (FG) bredigite (BR) into polyhydroxybutyrate-co-3-hydroxyvalerate and found that the osteogenic differentiation and mineralization of cells was enhanced by adding BR. The addition of FR and BR improved hydrophilicity and increased hydrolytic degradation. However, the Young’s modulus and the ultimate strength were reduced by the incorporation of FG and strengthened by further the incorporation of BR nanoparticles [76]. In the mussel-inspired method of Wang et al., a poly-L-lactic acid (PLLA) membrane was treated with dopamine to form polydopamine (PDA) coated PLLA, and PLLA was then coated with AgNPs using the reduction effect of PDA, which imparted antimicrobial activity to the modified membrane [77].

In addition, some other additives also enhance the performance of the membrane. PCLF is a derivative of PCL, with similar characteristics and structures. Ahmadi et al., by adding silicon and magnesium with fluorapatite nanoparticles to a PCLF/gelatin composite membrane, created a complex network in which various components interacted with each other to boost membrane performance. Membranes with 5% Si-Mg-FA nanoparticles showed suitable biological qualities and a reasonable degradation rate, with a 1.5-fold increase in mechanical properties [78].

##### 3.1.4.3. Addition of Nanoparticles

Many studies have mentioned that modifications in the form of nanoparticles can enhance the properties of raw materials for biocompatibility, bone regeneration properties, and mechanical strength [79]. Nanoparticles can improve the surface roughness of membranes and facilitate cell attachment and growth. Ye et al. found that a higher electrospinning solution density increased the precipitation of Sr-CAP nanoparticles and increased the surface roughness of PCL/chitosan membranes [80]. In addition, to combat the drawbacks of the low bioactivity of PCL, Jang et al. utilized the nano grooves to facilitate cell elongation [43]. The change of membrane mechanical strength depends on the distribution of nanoparticles, fiber diameter, and the interaction between nanoparticles and polymers [78]. R. Socrates found that by increasing the concentration of silver nanoparticles added to collagen-hydroxyapatite membranes, the mineralization increased, thus enhancing the mechanical properties [81]. In the study by José, ternary bioactive glass nanoparticles (BGNPs) were also shown to induce mineralization [82]. However, in some cases, the addition of nanoparticles can also cause a weakening of the mechanical properties, for example, crosslinking alginate with high amounts of nanohydroxyapatite can lead to a weakening of membrane plasticity [83]. Tahmineh et al. also found that the incorporation of either 5 or 10 wt% silicon and magnesium co-doped fluorapatite nanoparticles reduced the mechanical strength of PCLF/gelatin membranes, likely due to stress concentration caused by nanoparticle aggregation [78].

Moreover, the addition of nanoparticles can change the chemical activity of the membranes. Gheorghe found that TiO_2_ nanoparticles in polysulfone-silica microfiber composite membranes offered increased chemical resistance to acid oxidizers and bases [84]. TiO_2_ nanoparticles also have a photocatalytic effect and can produce reactive oxygen species by UV light irradiation, resulting in some antibacterial properties [84]. In addition, due to the high surface area to volume ratio of membranes co-blended with nanoparticles, they are able to carry and release drugs. In a study by Meifei et al., mesoporous silica nanoparticles (MSNs) loaded with dexamethasone can achieve targeted delivery and thus improve osteogenicity [79]. The addition of some antimicrobial nanoparticles, such as AgNPs, in membranes also leads to enhanced antimicrobial action due to their ability to expand the surface area [85].

### 3.2. Resorbable Membranes Based on a Natural Polymer

#### 3.2.1. Collagen

Collagen is the most widely used natural material in the creation of GTR/GBR membranes and, owing to its excellent biocompatibility and facilitation of wound healing and GBR, researchers have explored its application in the field of oral medicine [86]. Based on its absorbability, low immunogenicity, and ability to carry medicinal agents, among other factors, clinical results were nearly equivalent to those of nonabsorbable membranes [87]. Nevertheless, collagen itself is not mechanically suitable, since it lacks rigidity. It is more appropriate for applications in alveolar bone, such as for use in bone dehiscence and bone fenestration, where it can maintain stability without additional fixation. Collagen is also not particularly suitable for in vitro treatment and long-term cell culture. Additionally, it shares a common problem with absorbable membranes in that it does not offer space maintenance and it has a short degradation time [88].

Different experiments have been conducted to enhance the properties of collagen membranes. Simple collagen molecules are unstable so, in nature, they typically have a three-screw structure known as collagen fibrils, and these fibrils are arranged together to form collagen fibers by covalent cross-linking. Collagen is divided into various types, depending on its location and features. Among more than 20 kinds of collagen, type I collagen has a similar composition to periodontal connective tissue and is the main component for 90% of commercial collagen membranes (CM) [89].

The pure natural collagen membrane degrades relatively quickly, which means that it allows little support for bone regeneration. By contrast, the cross-linking of collagen membranes built by ultraviolet irradiation or chemical solution (e.g., glutaraldehyde) immersion, among other methods, shows better thermal stability and mechanical strength, effectively resisting the decomposition of collagenase solution for up to 50 days [90]. Research by Hong et al. has shown that by adding BCP (BCP, a mixture of HA and β-TCP) after UV cross-linking, the modified crosslinked collagen membrane has a greater ability to enhance neonatal bone formation and shows high biocompatibility and degradability [91]. Chia-Lai et al. reported that, after the monomer collagen was reconstituted into collagen fibrils, GLYMATRIX technology was used to glycosylate it with ribose, and ethylene oxide was used for sterilization (ribose cross-linking technology), to enhance the barrier function of the membrane [92].

Other attempts have been made to decelerate the degradation rate of the membrane. For instance, collagen membranes can be immersed and activated in autologous plasma rich in growth factors (PRGF) to add growth factors, enhancing and accelerating bone regeneration and predictable soft-tissue growth. The results revealed that the ultrastructure of three commercial collagen membranes showed excellent development in different directions, and their degradation rate slowed down at varying degrees [93]. Li et al. found that by adding 10% proanthocyanidins (OPCs) to the cross-linking agent of oligomeric OPCs in different concentrations, the OPCs-col membrane formed by type I collagen had the best overall performance and delayed the degradation of the membrane [90]. Using the gas-phase treatment of atomic layer deposition (ALD), Choy et al. introduced metal suture bonds, and collagen with Ti suture bonds was prepared by using TiO, which delayed the biodegradation time and enhanced the mechanical stability of the membrane [94].

In addition to extending the degradation time, the antibacterial performance of the collagen membrane has also been a focal point for collagen membrane improvement. In the process of GTR/GBR, it is easy to expose the membrane, and resulting infection is a significant problem existing in the clinical application of GTR/GBR membrane technology. Membrane exposure and infection have been reported in many clinical studies. Tempro and Nalbandian found that 1–2 weeks after surgery, the membrane was exposed, even though the soft tissue flap was completely covered [95]. Therefore, good antibacterial performance can reduce the infection caused by membrane exposure while promoting osteogenesis and soft tissue healing.

Socrates et al. mixed diverse percentages of spherical silver nanoparticles (tAgNPs) with collagen at 4 °C and added biomimetic HA composites for mineralization. The modified membrane showed excellent antibacterial properties and was suitable as a bone repair material that promoted calcified tissue repair potential [81].

Tovar et al. have explored a safer and more feasible method of sterilization. Commercially available acellular/degreased porcine pericardial collagen membranes were treated with supercritical CO_2_ (scCO_2_). The thickness of the membranes increased significantly, but the quality of the membranes did not change after treatment. Considering that scCO_2_ sterilization is a cold process, the possibility of changing the structure of the collagen material is reduced. Compared with traditional sterilization with radiation or acid/alkali solutions, which easily degrades the collagen structure, scCO_2_ treatment is a more reliable sterilization method [96].

Many efforts have been made to promote cell and bone differentiation, improve bone quality, and enhance other properties directly affecting bone regeneration. Gou et al. improved the mechanical properties (creating thicker and more organized fibers) and the hydrophilicity of membranes by cross-linking epigallocatechin-3-gallate (EGCG), promoting cell adhesion and osteogenic differentiation while maintaining good cellular compatibility [97]. Cho et al. prepared a collagen sponge (CS) using a 1% (*w*/*v*) collagen solution and added 0.1% (*w*/*v*) alendronate (ALN) to prepare an ALN-loaded collagen sponge (ALN-CS) as a carrier of growth factor rhBMP-2. Comparing CS, ALN-CS, CS containing rhBMP-2, and ALN-CS containing rhBMP-2, they found that the ALN-CS-rhBMP-2 group effectively inhibited bone resorption and promoted bone marrow formation, which was beneficial to the long-term and continuous improvement of bone quality [98].

Membrane improvement is mainly driven by the enhancement of barrier membrane function, the convenience of operation, and the ease of use. Various modifications have been attempted to improve the effectiveness of collagen membrane materials in alveolar bone augmentation and implantation, with varying degrees of effectiveness. Whether these modified membranes would be effective in clinical settings remains to be seen.

#### 3.2.2. Chitosan

Chitosan is employed in guided tissue regeneration as a linear polysaccharide containing glucosamine and N-acetyl glucosamine units linked by B-1,4 glycosidic bonds and is only dissolved in acidic solutions [99,100,101]. Due to its low cost and excellent immunogenicity, biocompatibility, biodegradability, and natural bacteriostatic fungal-suppressive characteristics, chitosan is commonly used in biomedical research. Many studies have incorporated chitosan with other polymers to strengthen the mechanical functions and bioactivity of scaffolds [102].

In addition to the characteristics already mentioned, chitosan shows low mechanical strength and a fast degradation rate. Fixed HA was mixed with CS to create a complex membrane, and the effect of the additive load on the membrane properties was studied. SEM analysis revealed that the surface of the composite membrane was both smooth and rough, with the roughness increasing as the HA content increased. After two months of culture, the degradation rates of the membranes were less than 22% of the initial weight, and they decreased with an increase in HA. These findings imply that HA derived from chicken can be formulated as an osteogenic filler to improve and regulate the bio-properties and degradation behavior of CS membranes, thereby guiding bone regeneration [103]. Su and his colleague are currently working on developing a GBR membrane made of PCL, gelatin, and chitosan that has been improved with β-tricalcium phosphate (β-TCP) for enhanced biocompatibility, mechanical properties, and antibacterial ability. Additionally, the results of electrospinning a chitosan-elastin solution to improve the mechanical properties of chitosan-based GBR membranes were presented in this study. Chitosan membranes containing elastin exhibited thicker fiber diameters, higher hydrophilicity, faster degradation rates, and higher mechanical strengths than chitosan membranes alone [104]. A layer of solid chitosan membrane and electrospun collagen nanofibers was used to create another type of membrane [105]. According to the biological and mechanical properties of collagen, the chitosan–collagen composite material can change the performance of the material when compared to chitosan or collagen alone [106].

Vale and his colleague developed a new type of antibacterial free-standing film by combining chitosan with hyaluronic acid (HAIB) to promote bactericidal and bioactive properties. Silver-doped bioglass nanoparticles (AgBGs) can be additionally combined to promote antibacterial and bioactivity properties [107] (Figure 7). Using electrospinning, they created a CS-AgNP/polyurethane composite (AgCSP) nanofiber membrane. Employing the ELS approach, they created a natural polymer membrane with a high antibacterial effect. The level of biocompatibility was adjusted by adding a reasonable amount of AgNPs. The AgNPs inside the membrane effectively promoted antibacterial activity. This membrane can be employed as a medical dressing material [85].

In an ideal GTR/GBR multilayer, one layer should sustain osteoblast attachment and proliferation, and the other layers should restrain cell adhesion. Therefore, mixing chitosan with other materials in multilayer structures is a promising technique to consider. Based on this idea, Vale et al. developed a novel antibacterial self-supporting membrane using natural polymer chitosan and hyaluronic acid in which AgBGs were incorporated to boost bactericidal and bioactive properties [107]. Another method consisted of developing a novel three-layer graded chitosan membrane (FGM) with bioactive glass gradients: BG and Pluronic F127 were joined inside of the membrane using a combination of the electrochemical and freeze-drying methods. Each layer provides an independent surface function and acts as a guide for the GTR membrane, resulting in a porous structure on the underside of the three-layer chitosan membrane that facilitates bone regeneration and also prevents bacterial entry [108].

As a high molecular polymer, chitosan has been widely studied and applied for use in GTR/GBR membranes due to its specific biological properties. It has immense potential, but because of its low mechanical strength, further research is needed to ensure its success in clinical practice.

#### 3.2.3. Gelatin

In the reconstruction of oral tissue and facial surgery, the hydrogel membrane is often used as a physical barrier [109]. Natural water-soluble polymers, such as gelatin, collagen, chitosan, hyaluronic acid, and synthetic polymers, especially PGA, PLA, and PCL, can be obtained from the thermal denaturation of collagen [110,111,112].

Compared with collagen, gelatin shows greater biocompatibility, biodegradability, lower cost with high efficacy, low immunogenicity, and good bioavailability. Therefore, gelatin is suitable for cell attachment, growth, and the maintenance of physiological functions. Developed in a previous study, an absorbable polycaprolactone-polyethylene glycol-polycaprolactone (PCEC)/gelatin-bismuth doped bioglass-graphene oxide bilayer showed superior mechanical, biochemical, and biological properties [113].

Due to the special location of the membrane, the gelatin-hyaluronic acid membrane was crosslinked with gelation and added with hydrogel to improve antibacterial performance and inhibit microbial contamination in the regeneration of soft tissue during potential acute inflammation [109]. In general, protein membranes and gel are indispensable in periodontal treatments.

#### 3.2.4. Silk Fibroin

Some membranes made from natural materials have been observed to have architectural and inflexible limitations, variable rates of degradation, and limited immune reactivity, preventing their clinical application. The silk fibroin membrane, a natural polymer and protein membrane, can be produced at a low cost and also has suitable mechanical properties with proven biocompatibility. Compared with the collagen membrane, the silk fibroin membrane promoted a greater amount of bone regeneration [114].

Although easy to produce, it is usually brittle under dry conditions, resulting in upper tensile strength, but low prolongation at breaking point [109]. One method to improve the elongation power, flexibility, and operability of the membrane is to add a plasticizer to the polymer resolvent. Here, the mixture of the silk fibroin protein, glycerin, and polyvinyl alcohol plasticizer can be verified by low-temperature thermal annealing coupled with a feasible and ordinary stabilization process. Compared with pure silk fibroin membranes, blends showed increased ductility, hydrophilicity, and subsequent proteolytic degradation.

Using this method, fibroin mixtures can be designed with customized mechanical, physicochemical, and biological properties for use in in vivo experiments and in the rehabilitation of injuries related to moderate periodontitis [114].

### 3.3. Magnesium Metal

We may deduce from the preceding study that magnesium and its alloys play an essential role in GTR/GBR. Coated magnesium or its compounds and surface modifications can limit biodegradability and enhance material properties.

There are many varieties of biocompatible membranes, although absorbable membranes are preferred because they do not require secondary operations. Metal membranes, in particular, outperform polymer composites and organic ceramic–polymer materials in terms of mechanical strength and structural integrity [115]. Magnesium, an absorbable metal membrane material, offers a variety of applications due to its excellent mechanical properties [116]. Magnesium is an important mineral used in the circulatory system of mammals and humans [117], with a daily recommended dose of 250–300 mg [118]. Magnesium also exhibits similar mechanical properties to those of human bone (e.g., elastic modulus, density, tensile strength) [117]. Scaffolds, bone plates, and wound closure devices are just a few of its modern medicinal applications [115,119,120]. Detailed in vivo experimentation using magnesium alloy implants within bone was described by Zheng et al., who demonstrated the osteoconductive potential of these implants [121]. However, the use of magnesium is limited by its rapid rate of degradation, especially in human fluids, due to pH and ion type in the physiological environment [122,123]. Scientists have tested a range of membranes deposited on the surface of magnesium metal and its compounds, as well as different procedures to modify these compounds, such as adding calcium, but no effective modification has yet been developed [124].

Magnesium can now be encapsulated using a variety of methods. Chitosan or collagen membranes, for example, can be destroyed while the magnesium core retains a certain mechanical strength. Physical vapor deposition (PVD)-coated materials can also be used to passivate magnesium. Jang et al. achieved the passivation of magnesium using plasma electrolytic oxidation and hydrothermal treatment, reducing the degradation rate by forming a dense layer of magnesium hydroxide containing phosphorus and calcium on the surface, maintaining the inner magnesium layer’s mechanical properties and biocompatibility [125]. Because magnesium materials can be used in a 100% biological water environment, Barbeck et al. used hydrofluoric acid to process the magnesium network and form magnesium fluoride [126]. The modified HF-Mg mesh was embedded into a collagen membrane to form a novel GTR/GBR membrane that has a lower degradation rate and good biocompatibility and mechanical strength. Ion-injection PVD coating is a new method to create a GTR/GBR membrane, although compared with non-PVD-coated membranes, this method does not provide a significant improvement; compared with the actual production and application of the membrane, pure magnesium is more suitable for GTR/GBR [127]. Furthermore, the use of chitosan coating can retard the rate of deterioration of magnesium. The composite chitosan–magnesium membrane (CS-Mg membrane) shows considerable potential for development as a conductive bone regeneration membrane with good osteogenic activity [128]. Lin et al. devised a novel ecologically friendly magnesium casting technology, which provides a broad platform for future application of magnesium materials [129]. To eliminate the significant amount of greenhouse gases produced in the magnesium casting process using sulfur hexafluoride (SF6), Lin et al. combined solid-solution heat treatment with surface treatment, replacing the Mg-5Zn-0.5Zr (ECO505) alloy with a new degradable and sustainable oral material. Du et al. created a magnesium-reinforced PLA polymer membrane, which showed good biocompatibility in animal experiments. Generally, magnesium contributes to mechanical properties [130]. Zhang et al. innovatively applied a magnesium-reinforced PLA membrane to the site of a bone defect. The membrane was made of a double-layer PLA membrane and a fluoride-covered AZ91 (9 wt% Al, 1 wt% Zn) (faz91) magnesium alloy core. After immersing the membrane in Hank’s Balanced Salt Solution (HbSS), among other conducted experiments, the researchers found that adding faz91 to PLA was biosafe, specifically for human use. This bioabsorbable PLA composite film can replace the traditional pure PLA film [60].

Magnesium and its alloy materials have been modified to provide improved performance and can be employed in the field of GTR/GBR. The methods described above lend substantial support for the utilization of magnesium materials in dentistry.

## 4. Clinical Trials & Future Research

The most important characteristics of chitosan materials for dental treatment are their biodegradability, biocompatibility, hydrophilicity, biological activity, and their antibacterial and antibacterial properties. In the past few decades, chitosan and its derivatives have been widely used in mouthwash, toothpaste, varnish, denture adhesive gel, dental pulp sealant, glass ion repair materials, and root canal sealant. In recent studies, chitosan biomaterials have also been used as titanium implant coatings, dental membranes, stents, hemostatic dressings, and carriers for drug or gene delivery. According to the success of the treatments, chitosan has been widely used in various dental applications. However, more research is needed to further determine the properties of chitosan biomaterials and expand their effective use in dental treatment. Hydrogels are polymer networks composed of cross-linked hydrophilic chains [131]. With a high affinity for water’s physical properties, water gel allows excessive integration with the surrounding tissue, reduces the possibility of the inflammatory response from the natural water gel to the synthetic hydrogels, allows for its application as a bioactive molecule carrier, and is ell suited for additive manufacturing technology; deeper research is required into the application of natural water gel [132].

The most important reason why magnesium material can be widely used in dentalclinics is the good mechanical properties due to its degradability and biocompatibility. By using different methods to modify it, its degradation rate can be reduced, allowing for wider clinical use. Scientists have used an environmentally friendly way to cast a degradable and regenerative film [129]. As a traditionally popular GTR GBR film, collagen has many advantages while being less mechanically stable. It is now clinically possible to glycosylate and sterilize this film using ethylene oxide for enhanced mechanical purposes during the disinfection step [92]. In addition, the introduction of metal seams can improve the mechanical properties while reducing the degradation rate [94]. In the future, the traditional collagen membrane may become even more valuable through the use of more scientific research results. Similar to collagen films, PLA (Polylactic acid) has relatively low mechanical properties. Clinically, PLA tissue regeneration scaffolds are often prepared by electrospinning, providing a greater application potential in a three-dimensional space, and their porosity can be accurately measured using 3D-printing technology. As a new synthetic membrane, PLA has greatly reduced clinical application costs due to its renewability. In short, PLA offers great potential in the future. Polylactic-co-glycolic acid (PLGA) shows similar bioresorption to collagen. Clinically, additional factors may be added to modify its cell permeability. [65]. However, the modified PLGA membrane is still in the experimental stage and is currently only used for in vitro and in vivo studies in animals; in the future, PLGA may be used in the clinic as a unique GTR/GBR membrane to improve cell activity and antimicrobiality.

## 5. Conclusions

The results of the above studies generally described moderate to substantial advances in the modification of degradable membranes for GTR/GBR. For various biodegradable materials, the commonly applied modification methods include: (1) the addition of substances to enhance membrane performance, such as proteins and growth factors for bone and tissue regeneration; nanoparticles that can enhance several properties of the membrane; components that can release silicate, calcium ions, and phosphate to enhance osteoconductivity and bone formation; and plasticizers that improve the elongation ability of silk fibroin membranes; (2) the application of novel preparation methods or a combination of various existing preparation methods such as, the use of 3D printing, air plasma, the combination of electrostatic spinning and photo-crosslinking, the combination of ultraviolet irradiation or chemical solution immersion and cross-linking, to enhance multiple membrane properties.

The formation of composite membranes from different materials is a new and growing trend and can be applied to overcome the deficiencies of the respective materials. The identification of suitable combinations of modification methods and materials should remain the focus of future research. In addition, in vivo experiments, as well as clinical studies, should apply promising modification methods following extensive risk assessment.

## Figures and Tables

**Figure 1 polymers-14-00871-f001:**
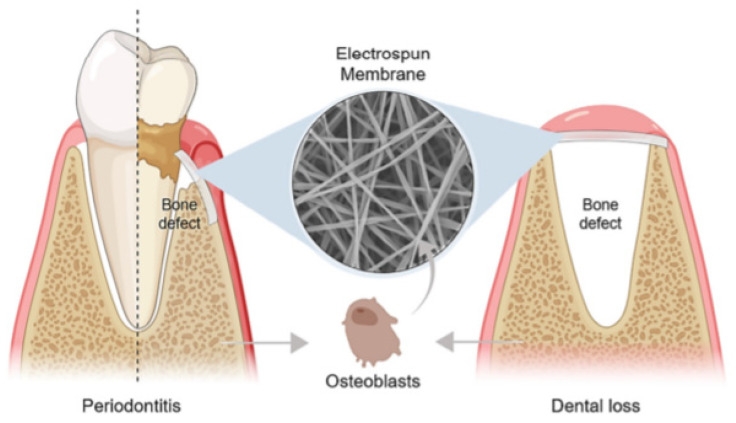
The application of an electrospun membrane for guided tissue/bone regeneration (GTR/GBR) in periodontitis and dental loss, respectively. Reprinted from Ref. [10].

**Figure 2 polymers-14-00871-f002:**
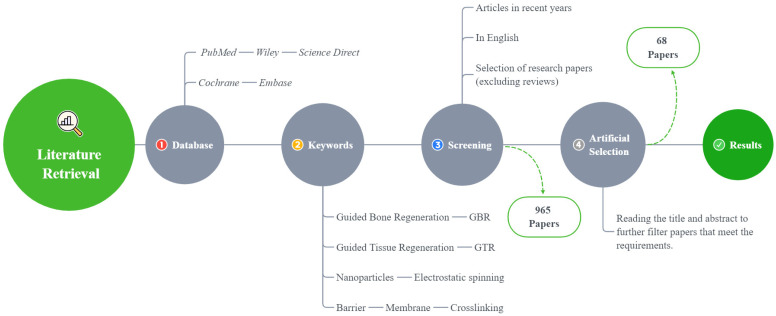
A flowchart of the literature search method.

**Figure 3 polymers-14-00871-f003:**
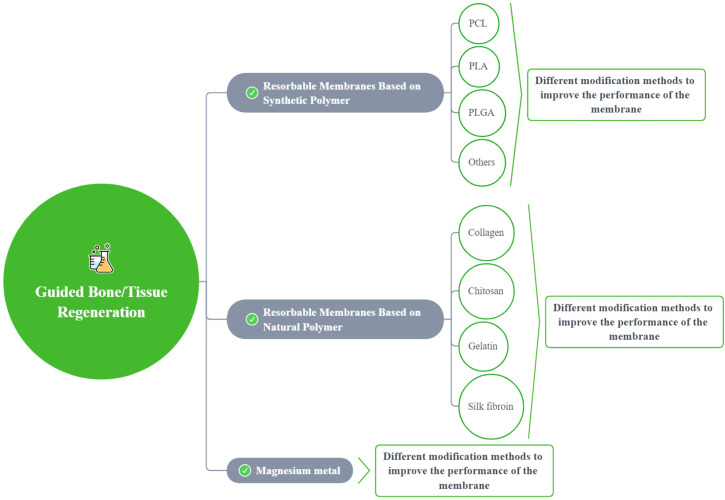
A sketch showing the modification methods for different membranes.

**Figure 4 polymers-14-00871-f004:**
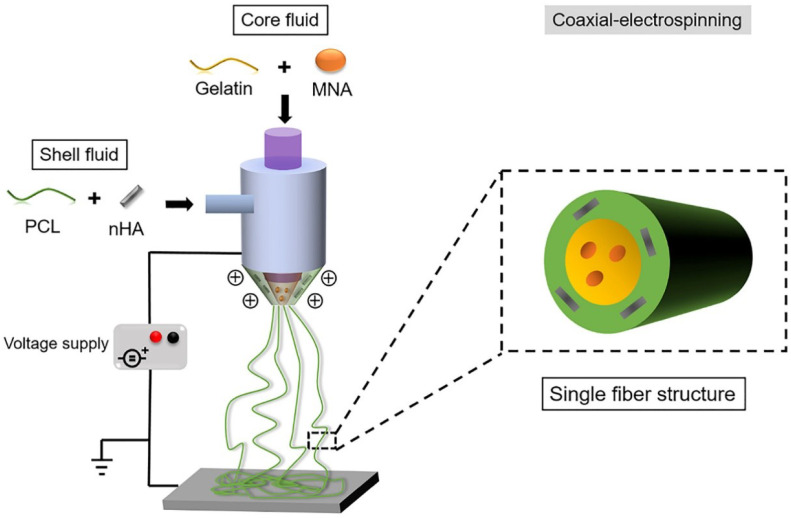
Generating the core-shell structure, reprinted from Ref. [41].

**Figure 5 polymers-14-00871-f005:**
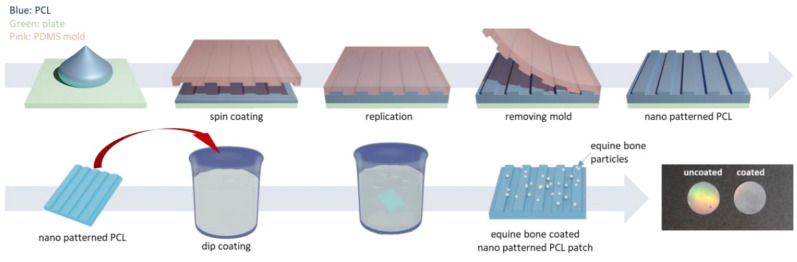
The preparation of equine bone powder (EBP)-coated polycaprolactone (PCL) nano-patterned patches, reprinted from Ref. [43].

**Figure 6 polymers-14-00871-f006:**
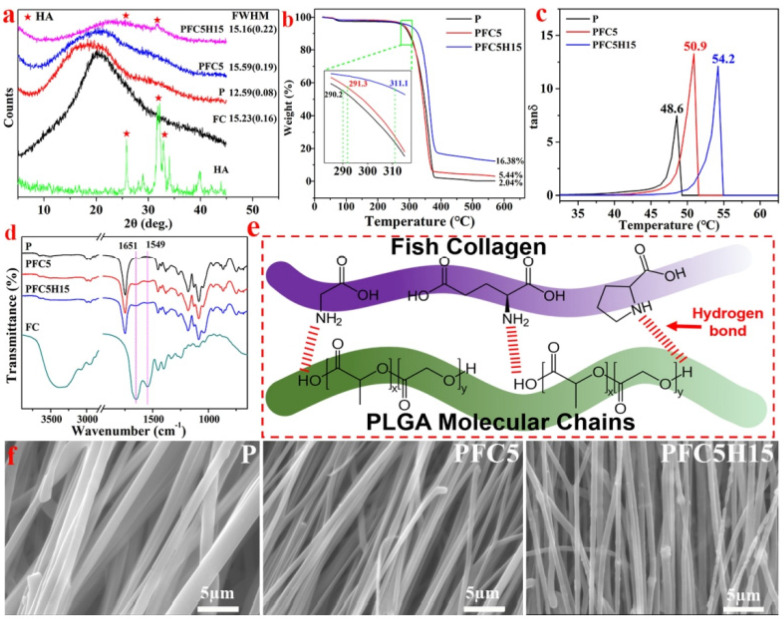
(**a**) The X-ray diffraction (XRD) patterns of HA, FC, and three different membranes; (**b**) the TG curve and (**c**) the DMA curve of the membranes illustrating the characteristics; (**d**) the FTIR spectra and (**e**) a schematic diagram of the P, PFC5, and PFC5H15 membranes and FC; (**f**) the morphology of the edge of a fiber fracture, reprinted from Ref. [69].

**Figure 7 polymers-14-00871-f007:**
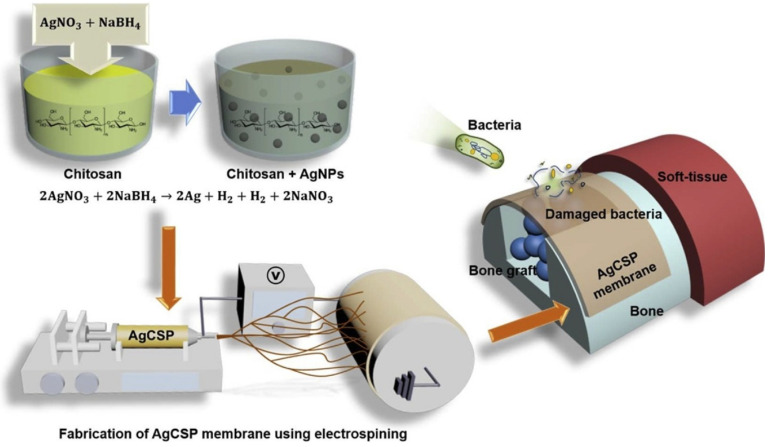
The generation of the AgCSP membrane and its application, reprinted from Ref. [85].

**Table 1 polymers-14-00871-t001:** The various properties of modified PCL membranes.

Year [Ref.]	Main Membrane Material	Modifications	Additional Properties	Drawbacks
2019 [32]	MNA, PCL, polydopamine	Coated with polydopamine and the addition of MNA	Controlled MNA release for antibacterial activity	Not mentioned
2021 [33]	Polylactic acid (PLA)/cellulose acetate (CA) or PCL, AgNPs, hydroxyapatite nanoparticles (HANPs)	Adding AgNPs, HANPs	Sustained antibacterial activity, optimized mechanical properties, lowered degradation rate, enhanced cell proliferation	HANPs: 20 wt%, decreased tensile property
2018 [34]	PCL, PEG, bioactive glass (BGs)	Adding BGs	Suitable mechanical and biodegradable properties, hydrophilic surface, higher proliferation rates of adipose-derived stem cells, good bone mineralization capacity	Not mentioned
2018 [36]	F18 bioactive glass, PCL	Adding F18 bioactive glass	Enhanced osteogenesis and excellent tensile strength	Not mentioned
2017 [37]	Si-NPs, PCL	Adding Si-NPs	Improved mechanical properties	Not mentioned
2019 [39]	SiO_2_-CaO-P_2_O_5_ and SiO_2_-SrO-P_2_O_5_, bisphosphonate drug ibandronate, PCL	Two different types of mesoporous bioactive glasses, bisphosphonate drug ibandronate	Bioactive glass enhanced hydrophilicity and bioactivity; Sr^+^ bisphosphonate drug ibandronate improved osteogenesis	Not mentioned
2015 [40]	PCL, ZnO	Adding ZnO	Antibacterial properties, enhanced cell proliferation/wound healing	Decreased mechanical suitability after adding ZnO; adding 30 wt% ZnO decreased viability
2018 [41]	metronidazole (MNA), nano-hydroxyapatite (NHA), PCL, gelatin	Adding MNA, NHA, forming core-shell structure	Promoted osteogenesis and slow MNA release for antibacterial activity	Not mentioned
2018 [42]	PCL, NHA/BG	Adding NHA/BG	Enhanced mechanical properties, excellent cell attachment	The membrane with a high nHA/BG loading density was pooer than the low one
2020 [43]	EBPs, PCL, hydroxyapatite (HA)	Forming nanopattern and the addition of EBPs	EBPs enhanced surface hydrophilicity; nanopattern and EBPs enhanced the osteogenic phenotype of human dental pulp stem cells (DPSCs)	Not mentioned
2019 [44]	PCL, Strontium-substituted hydroxyapatite nanofibers (SrHANFs)	Adding SrHANFs	Promoted differentiation and mineralization of osteoblast-like cells	Not mentioned
2019 [45]	PCL PolyHIPE	Air plasma treatment	PCL PolyHIPE layer promoted osteogenesis, Ca and mineral deposition of bone cells, the deposition of collagen; electrospun nanofibrous PCL layer promoted cell-occlusion	Not mentioned
2018 [46]	BG, PCL	Adding BG	Excellent mechanical properties	Not mentioned
2016 [47]	PCL, bioactive CaO-SiO_2_,	Hydroxyapatite-coated	Osteoconductivity and excellent bone formation ability	Not mentioned
2018 [48]	PCL, gelatin, chitosan, β-tricalcium phosphate (β-TCP)	Adding β-TCP	Enhanced osteogenesis, adjustable degradation rate, more wettable surface, suitable mechanical properties	Not mentioned

## Data Availability

All data, figures, and tables in this review paper are labeled with references.

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
