# Peer review of "Advances in Modification Methods Based on Biodegradable Membranes in Guided Bone/Tissue Regeneration: A Review"

_polymers, 2022, doi:10.3390/polym14050871_

Round 1

Reviewer 1 Report

Dear authors,

I have read the review, 'Advances in Modification Methods Based on Biodegradable Membranes in Guided Bone/Tissue Regeneration: A Review' with high interest. The authors have summarized the works nicely and provided quality images to support the work. Hence I am recommending to accept this review after minor revision.

  1. Please separately write a section to mention the current clinical trials.
  2. What are the current challenges and what future research can be explored?

Author Response

Q 1)

Please separately write a section to mention the current clinical trials.

A 1)

Thanks very much for your suggestion. Based on your suggestion, we have added the necessary content to 4. Clinical trials & Future research. (Line 692-730)

Q 2)

What are the current challenges and what future research can be explored?

A 2)

Thank you very much for your question. The current challenges and future research were added to the new section mentioned above. (Line 692-730)

Thank you once again for your comments, which significantly improved the quality of this manuscript.

Reviewer 2 Report

The authors reviewed the recent advance in membrane technologies in guided bone regeneration. The authors mentioned they derived the evidence from the latest 5 years, however, it is unclear what is the selection protocol. Overall, it is an informative paper but I feel there is a lack of synthesis and quality appraisal of the evidence included. Perhaps the authors can add a perspective or future direction section to strengthen those aspects. Other minor comments include the following: Figures: Are all the figures reused with permission?  Line 62-67 should be referenced.  Table 1: For drawbacks, if they are not mentioned by the original authors, I think it should be indicated as "not mentioned", rather than a "-".  Line 280: "... improved by o the fabrication process..." delete o Figure 4: please indicate the source of this figure. Line 416-433 should be referenced. Line 617: "Magnesium" should not be capitalised; Line 634: "Created" should not be capitalised. 

Author Response

Q 1)

Perhaps the authors can add a perspective or future direction section to strengthen those aspects.

A 1)

Thank you very much for your comments. The current challenges and future research were added to a new section, 4. Clinical trials & Future research. (Line 692-730)

Q 2)

Other minor comments include the following:

2a) Figures: Are all the figures reused with permission?

2b) Line 62-67 should be referenced.

Figure 4: please indicate the source of this figure.

Line 416-433 should be referenced.

A 2)

2a) Thank you very much for your question. We have learned that all the figures were reused with permission. And all licenses will be sent in a PDF file as an attachment, please have a look.

2b) Based on your detailed suggestions, we have made changes to the article.

In line 62-67, the standard we mentioned is ISO 22803.

We have indicated the reference in the figure notes of Figure 6 (Figure 4 of the original manuscript). (Line 349-353)

We have indicated the reference in Line 462-472 (Line 416-433 of the original manuscript).

Q 3)

Other minor comments include the following:

Table 1: For drawbacks, if they are not mentioned by the original authors, I think it should be indicated as "not mentioned", rather than a "-".  Line 280: "... improved by o the fabrication process..." delete o.

Line 617: "Magnesium" should not be capitalised;

Line 634: "Created" should not be capitalised.

A 3)

Thank you very much for your advice. Based on your detailed suggestions, we have made changes to the article. The modifications are in Line 292, Line 663 and Line 680.

Thank you once again for your comments, which significantly improved the quality of this manuscript.

Reviewer 3 Report

The review covers the main polymers and their modifications. In general, it is written in detail, but there are a few remarks for a clearer understanding of the topic.

The authors should provide an introductory sketch or scheme demonstrating the different types of polymers and methods  used for the guided Bone/Tissue regeneration.

I would ask the authors to provide the literature search method and its flowchart in the
manuscript.

In Figure 4, you must indicate the references on the sources of figures.
It would be nice to have a section that would describe all modifiers in the form of nanoparticles that enhance biocompatibility, mechanical strength and other properties of materials. Many studies, for example, are devoted to the introduction of halloysite clay nanotubes into polymers to create tissue engineering scaffolds (Polymers, 2021, 13(22), 3949, ACS Biomater. Sci. Eng. 2019, 5, 8, 4037–4047), as well as the addition of other nanomaterials of various types. This section is necessary, since in the introduction it was stated about changing the properties of scaffolds due to nanomaterials

Author Response

Q 1)

The authors should provide an introductory sketch or scheme demonstrating the different types of polymers and methods used for the guided Bone/Tissue regeneration.

A 1)

Thanks very much for your suggestion. Based on your suggestions, we have provided an introductory sketch of the modification methods for different membranes (Figure 3) in the 3rd section. (Page 5)

Q 2)

I would ask the authors to provide the literature search method and its flowchart in the manuscript.

A 2)

Thank you very much for your advice. We have added a new section, 2. Materials and Methods, and its flowchart to the text. (Line 150-154) The details are as follows:

In this review, only the literature published in English is included, and the last search was conducted in August 2021. A literature search was carried out in PubMed, Wiley, Cochrane, Embase, and Science Direct, and the published years were limited to nearly 5 years, including in vitro, in vivo and human studies, excluding reviews. Various improved methods of absorbable barrier membranes for GTR/GBR were expounded. It contains papers related to this topic published before August 2021. The following keywords are used in different combinations: “Guided Bone Regeneration” , “Guided Tissue Regeneration” , “GBR” , “GTR” , “Barrier” , “Membrane” , “Crosslinking” , “Electrostatic spinning” , “Nanoparticles”. A total of 965 articles were identified by electronic database and manual search. 68 articles met the inclusion criteria. The title and abstract of the study were independently selected by two reviewers and classified as suitable or unsuitable for inclusion. For studies that appear to fit topics or headings of interest and for which there is insufficient information to make a definitive decision, an independent review of all reports is conducted. After checking the references of the identified articles, a manual search was also carried out.

Q 3

3a) In Figure 4, you must indicate the references on the sources of figures.

3b) It would be nice to have a section that would describe all modifiers in the form of nanoparticles that enhance biocompatibility, mechanical strength and other properties of materials. Many studies, for example, are devoted to the introduction of halloysite clay nanotubes into polymers to create tissue engineering scaffolds (Polymers, 2021, 13(22), 3949, ACS Biomater. Sci. Eng. 2019, 5, 8, 4037–4047), as well as the addition of other nanomaterials of various types. This section is necessary, since in the introduction it was stated about changing the properties of scaffolds due to nanomaterials.

A 3)

3a) Thanks very much for your suggestion. Based on your suggestions, we have indicated its reference in the figure notes of Figure 6 (Figure 4 of the original manuscript). (Line 349-353)

3b) Thank you very much for your advice. We have added a new section, 2.4.3 Addition of nanoparticles, to describe modifiers in the form of nanoparticles. (Line 428-459)

Thank you once again for your comments, which significantly improved the quality of this manuscript.